# First Responder Resiliency ECHO: Innovative Telementoring during the COVID-19 Pandemic

**DOI:** 10.3390/ijerph18094900

**Published:** 2021-05-04

**Authors:** Joanna G. Katzman, Laura E. Tomedi, George Everly, Margaret Greenwood-Ericksen, Elizabeth Romero, Nils Rosenbaum, Jessica Medrano, Paige Menking, Gaelyn R.D. Archer, Chamron Martin, Karina A. Dow, Shannon McCoy-Hayes, Jeffrey W. Katzman

**Affiliations:** 1ECHO Institute, University of New Mexico, Albuquerque, NM 87131, USA; LTomedi@salud.unm.edu (L.E.T.); menking@ncfh.com (P.M.); grarcher@salud.unm.edu (G.R.D.A.); chamartin@salud.unm.edu (C.M.); KADow@salud.unm.edu (K.A.D.); smccoyhayes@salud.unm.edu (S.M.-H.); 2Department of Neurosurgery, University of New Mexico, Albuquerque, NM 87131, USA; 3The Johns Hopkins Bloomberg School of Public Health, Johns Hopkins University, Baltimore, MD 21205, USA; geverly1@jhmi.edu (G.E.); elaromero@salud.unm.edu (E.R.); 4Department of Emergency Medicine, University of New Mexico, Albuquerque, NM 87131, USA; MGreenwoodEricksen@salud.unm.edu; 5Behavioral Sciences Section, Albuquerque Police Department, Albuquerque, NM 87102, USA; nrosenbaum@cabq.gov; 6Emergency Medical Services Program, Central New Mexico Community College, Albuquerque, NM 87106, USA; jmiller72@cnm.edu; 7Department of Psychiatry and Behavioral Sciences, University of New Mexico, Albuquerque, NM 87131, USA; jekatzman@salud.unm.edu

**Keywords:** first responder, resiliency, self-care, COVID-19, healthcare worker, stress, law enforcement, paramedic, emergency medical technician

## Abstract

The First Responder ECHO (Extension for Community Outcomes) program was established in 2019 to provide education for first responders on self-care techniques and resiliency while establishing a community of practice to alleviate the enormous stress due to trauma and substance misuse in the community. When the SARS-CoV-2 (COVID-19) pandemic hit the United States (US) in March 2020, a tremendous strain was placed on first responders and healthcare workers, resulting in a program expansion to include stress mitigation strategies. From 31 March 2020, through 31 December 2020, 1530 unique first responders and frontline clinicians participated in the newly expanded First Responder Resiliency (FRR) ECHO. The robust curriculum included: psychological first aid, critical incident debriefing, moral distress, crisis management strategies, and self-care skills. Survey and focus group results demonstrated that, while overall stress levels did not decline, participants felt more confident using psychological first aid, managing and recognizing colleagues who needed mental health assistance, and taking time for self-care. Although first responders still face a higher level of stress as a result of their occupation, this FRR ECHO program improves stress management skills while providing weekly learning-listening sessions, social support, and a community of practice for all first responders.

## 1. Introduction

Society relies on frontline healthcare clinicians (non-traditional first responders: NTFRs) and traditional first responders, or TFRs (law enforcement, firefighters, and emergency medical technicians), to handle the most serious of extreme weather events, terror-related threats, and daily community health-related injury and illness. Traditional first responders are known to be at higher risk of developing posttraumatic stress disorder (PTSD), depression, suicide, anxiety, alcohol use disorder, and sleep disturbances compared to the general population [1]. In many rural communities, TFRs are at even higher risk of developing mental health diagnoses as a result of their stressful work environment [2]. Rural first responders also report high-stress work-related issues, such as a lack of resources and training, as factors that impact their sense of self-efficacy [3].

A plethora of research on the psychological needs of first responders in the United States (US) increased significantly in 2001 after the World Trade Center attacks in New York City [4]. This research identified the characteristics of resiliency in individuals, leaders, and organizations as well as the dire need for wellness programs, resilience training, critical incident debriefing, increased awareness about vicarious trauma, compassion fatigue, and moral injury [5]. Conclusions from the Institute of Medicine Report in 2012: *Building a Resilient Workforce: Opportunities for the Department of Homeland Security*, suggested that the characteristics of highly resilient individuals who have suffered psychological trauma usually include: “optimism or faith, integrity, social support, decisiveness, perseverance, and self-control.” George Everly, Ph.D., one of the primary authors, noted that the level of social support created the most variation in the level of resiliency [6]. Many first responder programs have demonstrated improved resiliency through a variety of psychological training. The HEROES Project promoted first responder self-efficacy and resiliency and demonstrated improvement in first responder mental health [7,8]. The COP-2-COP Hotline, originally established in 1999 to assist police officers and their families in New Jersey, was one of the earliest peer support hotlines and became very useful during the 9/11 tragedy [9]. The Road to Mental Health Readiness (R2MR), a four-hour Canadian training program with traditional first responders of all types (correction officers, emergency medical services, fire, and police), demonstrated a significant increase in resiliency skills and a reduction in the stigma of mental illness [10]. A seven-hour, in-person TFR resiliency program in Colorado recruited first responders who had involvement with the Aurora, Colorado mass shooting incident in 2012. The study demonstrated significant improvement in sustained resiliency scores [11].

More recently, the SARS-CoV-2 (COVID-19) pandemic has delivered unprecedented challenges and tested the moral fabric of healthcare workers and TFRs globally. Many TFRs and NTFRs on the frontlines are both physically and mentally fatigued, but relatively few have training in the importance of self-care or stress management [12]. Non-Traditional First Responders, including emergency room staff, intensive care unit clinicians, along with medical and behavioral health personnel, have little training on these topics and strategies [13]. This pandemic has exhausted healthcare workers through overwhelmed health systems, unprecedented mortality rates, shortages of personal protective equipment, and fear of virus transmission to self or family. In a cross-sectional study of 1257 healthcare workers in 34 hospitals throughout China, clinicians and others on the frontline caring for patients suffering from COVID-19 were found to have significantly increased rates of mental health symptoms, including depression, anxiety, sleep disturbance, and stress [14]. In terms of the high rates of stress to frontline healthcare workers and first responders, the COVID-19 pandemic is not unusual. Studies of prior pandemics, such as the 2003 Severe Acute Respiratory Syndrome (SARS) and the 2015 Middle Eastern Respirator Syndrome (MERS), found that healthcare workers showed higher rates of alcohol use, depression, anxiety, PTSD, and psychological stress, and these symptoms can last up to three years after the pandemic has ended [15,16,17].

Research further describes the relationship between social connection and resilience. Bzodik and Dunbar review studies correlating loneliness and social isolation with increased cardiovascular disease and earlier mortality, decreased immune response, and higher levels of depression. First responders often describe debriefing with one another at the end of a shift, though opportunities to do this amidst a pandemic have constricted with guidelines for social distancing. A program aimed at the development of resilience in first responders, particularly in a time of social isolation, must provide not only didactic information but also the opportunity for connection [18].

Realizing that frontline clinicians may benefit from resiliency training, the First Responder Resiliency (FRR) ECHO program was developed as an extension of Project ECHO’s existing First Responder program to address the psychological needs of all frontline clinicians, in addition to TFRs [19,20]. It has provided both an opportunity for learning critical information and skills as well as the opportunity for connection with one another through a listening group format.

### Brief History of Project ECHO

Project ECHO (Extension for Community Healthcare Outcomes) at the University of New Mexico (NM), is a novel telementoring model to synchronously connect primary care clinicians working in rural and urban underserved regions with specialty care clinicians who could support their practices with up-to-date information and case-based learning [21]. There are now ECHO programs for more than 70 different medical, public health, and education-related topics offered in every US state and over 44 different countries [22]. This article describes the FRR ECHO program curriculum while measuring its impact through surveys and focus groups from 23 March 2020, through 31 December 2020.

## 2. Materials and Methods

### 2.1. Program, Hub Team, and Demographics

The FRR ECHO began 23 March 2020, and with meetings scheduled weekly for sixty minutes [23]. This program is designed to bring frontline healthcare workers and first responders (e.g., participants) together in a synchronous, virtual forum to share best practices regarding self-care and evidence-based psychological stress mitigation strategies so participants can apply these tools in their work and home environments. Each session includes an evidence-based didactic presentation and a facilitated listening session. The listening session consists of the participants breaking out into small groups for 15–20 min whereby a FRR ECHO hub team member facilitates a discussion related to the presentation for the particular session and an opportunity to connect with one another through the discussion of shared experiences.

First Responder Resiliency ECHO sessions are Zoom-based (Zoom, Inc., San Jose, CA, USA), which tracks participant attendance. The FRR ECHO hub team includes three psychiatrists, a neurologist, an emergency department physician, an emergency medical technician (EMT), a psychologist, and a community health worker (CHW). The hub team is an ECHO concept which describes the subject matter experts. One psychiatrist leads the state’s largest psychiatric emergency room, while the other is employed by the state’s largest police department. Programmatic and administrative support is provided by dedicated ECHO staff members. Guest presenters are identified as experts in the fields of resiliency training, stress management, critical incident debriefing, and trauma counseling.

Participants register prior to the session by entering their name, email, organization, location, license, and first responder type (EMT fire, law enforcement, other). Other demographics, such as gender, race/ethnicity, and age are not collected regularly as a part of registration. Project ECHO staff members, hub team, guest presenters, and participants who attended less than 10 minutes of any session were excluded from the analysis. Attendance counts included both telephone and video. License and first responder type were combined to create a profession variable: traditional first responder (TFR), medical clinician (MD, DO, nurse practitioner, physician assistant, nurse), mental health professional, and other (Ph.D., educator, pastor/rabbi, etc.). No-cost continuing education credits were provided to all disciplines except dentists.

### 2.2. Curriculum

The curriculum focused on providing psychological tools and self-care strategies for frontline clinicians. Topics included psychological first aid, critical incident debriefing, burnout, compassion fatigue, moral injury, mitigation strategies, and descriptions of peer support programs. Additional sessions addressed the hospitalized patient experience and COVID’s impact on other public health crises, including substance use disorder and chronic pain. Guest speakers have included the University of Minnesota (“Battle Buddies”), Emory University (“Community Resiliency Model”), the University of Indiana (“Internal Peer Support”), and the Baltimore Police Department.

### 2.3. Participant Survey

An online retrospective pre-post survey using RedCap (Vanderbilt University, Nasville, TN, USA) was administered to measure the impact of the FRR ECHO curriculum [24]. The survey was developed by the program evaluators and staff using information gathered from a literature review, previous assessments among first responders, program objectives, and input from first responder stakeholders [25]. The hub team reviewed and piloted the survey. Survey topics included confidence in resiliency skills, evaluating work stressors, social support, and coping skills. Confidence was measured on a 4-point scale (1-Not at all confident, 2-Somewhat confident, 3-Moderately confident, 4-Extremely confident). Agreement was measured on a 5-point Likert scale (Strongly disagree, Disagree, Neutral, Agree, Strongly Agree). Mean scores were calculated for retrospective pre- (respondents were asked after they had already attended, but they were asked to think back to before they attended the ECHO) and post-responses and compared using a two-tailed t-test. Program staff sent the survey to any person who had attended at least once between September 2019–October 2020 (N = 1294).

### 2.4. Focus Groups

Three 60-min. focus groups were conducted virtually with invited participants who had attended three or more sessions. Participants were offered one of three different time slots (morning, afternoon, evening) to best accommodate their schedule. Discussions were recorded, transcribed, and then analyzed with Dedoose© [26]. Transcripts were coded independently by two staff, and consensus coding was reached. There were 18 participants in the focus groups occurring 5–9 October 2020. Focus group participants were categorized into two groups for analysis: rural (population < 50,000 people) versus urban (≥ 50,000 people) and traditional first responder (TFR) versus non-traditional first responder (NTFRs). Evaluation staff reviewed coding, sorted it into categories, and identified categories into major themes.

Focus group questions included:How has participation in the First Responder Resiliency ECHO program impacted you?
After open discussion, the focus group facilitator probed for impact on:Self-care or coping skills;Dealing with anxiety and stress;Listening to family members, colleagues, or patients;Quality of relationships.How has participation in this ECHO impacted the way you respond to or work in stressful environments?What challenges or barriers have you experienced to using the materials and skills from this ECHO?Do you have any suggestions on how to improve this ECHO or its content?

This study was approved by the Institutional Review Board (IRB) of the University of New Mexico Health Sciences Center (#20-045).

## 3. Results

### 3.1. Attendance

From 23 March 2020, through 31 December 2020, there were 34 FRR ECHO sessions. The program had 1530 unique participants (1369 on video and 161 by telephone). Attendance peaked in April (with 384 participants on 6 April 2020, directly after the COVID pandemic shutdown) and then stabilized to between 60 and 124 participants per session beginning in June. On average, 108 first responders attended each session. After the peak, an average of 94 participants attended each session. Participants attended 2.37 (range: 1–31) sessions on average. Although the majority of participants attended once, approximately 239 people attended regularly (≥4 sessions). See Figure 1.

Although Project ECHO does not collect gender information for each unique participant, the participant survey and the focus group demographics showed that about twice as many FRR ECHO evaluation participants were female. Of the 45 survey participants, 30 were female, while the focus group participants included 14 females and 4 males. Among participants who had registration data (N = 919), nearly one in four were TFRs. Non-traditional first responders represented over 75% of the participants. Job category was similar between all participants and those who participated in the evaluation (the survey and the focus groups). Over half of the survey and focus group respondents were female. See Table 1.

### 3.2. Participant Survey

Forty-five people responded, which was 3.5% of the total participants and 26.8% of the 168 people who attended FRR ECHO sessions when the survey was active (2 November–16 November 2020). Evaluation staff compared the distribution of licenses/professions and location among all participants that had registration data to survey respondents. No significant differences between participants and survey respondents were found.

Survey respondents reported increased self-efficacy with all the skills covered during sessions. Respondents reported they were more confident using psychological first aid, practicing self-care, recognizing and managing emergencies related to a mental illness, and incorporating trauma-informed care into their practice. Additionally, they felt significantly more confident in responding to a coworker suffering from mental health and/or substance use issues. Measures of work stress (e.g., fatigue, feeling overwhelmed, sense of danger, etc.), however, did not change for survey respondents. When asked how they felt now, respondents reported feeling fatigued, even when they had enough sleep. Respondents did not report a change in most coping mechanisms when asked to think back before they began attending the FRR ECHO. The two significant findings were: (1) participants demonstrated a decreased tendency to “shut down” in times of stress, and (2) participants endorsed drinking more alcohol than planned in order to deal with stress. See Table 2.

### 3.3. Focus Groups

In the focus groups, the participants were probed on their thoughts on the program. They shared thoughts including (1) the overwhelming nature of COVID sometimes prevented them from using the skills that they learned in the FRR ECHO, (2) the FRR ECHO provided a sense of community where they felt safe to share their stress and trauma, (3) they liked the content of the didactics during the FRR ECHO sessions. These observations were coded, and four major themes emerged from the codes. The four major themes identified by the focus group participants were: (1) external factors are a major cause of stress for participants, (2) the FRR ECHO sessions provide a sense of community, (3) and are high quality, and (4) participants are sharing the skills they learned in the FRR ECHO with others. Participants reported their biggest challenge to incorporating resiliency skills into their work and personal lives was coping with external factors, such as political turmoil and the overall stress of the pandemic. The pandemic left many first responders feeling like they were “serving in a battle” and expressed that vulnerability could make them seem weak to patients and coworkers. Participants also mentioned that they used the skills they learned. These concerns were predominantly seen in the NTFR and urban participants. Focus group participants of all types felt that the greatest benefit of the FRR ECHO is the community of practice. Specifically, they felt that the FRR ECHO normalized what they were feeling and that they had found a group of people they could trust. Additionally, they noted that they shared the materials they learned during the ECHO sessions. Participants, however, also expressed a need for more time to review the session content and would prefer to be placed in a listening group with first responders with similar experience levels. See Table 3 and Table 4.

## 4. Discussion

The FRR ECHO has successfully created a community of practice around the topic of resiliency for many TFRs and NTFRs during the COVID-19 pandemic. Many participants throughout the globe have joined these sessions twenty or more times, and have actively participated in the facilitated listening sessions during this virtual program, now one year later.

For the participants completing the survey and focus groups, there was a significant improvement in confidence related to understanding psychological first aid, practicing self-care, recognizing and managing emergencies related to a mental illness, and incorporating trauma-informed care into their practice. Participants in the focus group reported a feeling of increased trust and stated that the ECHO sessions helped to normalize their feelings. The First Responder Resiliency ECHO is providing a social support network deemed so critical to resiliency after the World Trade Center tragedy.

### 4.1. Weekly FRR ECHO Sessions

The COVID-19 pandemic has tested the physical, mental, and moral character of the TFR and NTFR communities, yet both historically have very little training in self-care and/or stress management [27]. The FRR ECHO provides evidence-based training on many aspects of self-care and resiliency, and the participants seem to find value in these weekly sessions. The average number of 94 weekly participants per FRR ECHO session has remained consistent throughout the last eleven months (April 2020–present). Unique FRR ECHO participants have averaged over two sessions each. Additionally, 239 of them have attended ≥4 sessions, 60 have attended ≥10 sessions, and 15 have attended ≥21 sessions. These are very high returning rates, not only compared to other successful Project ECHO programs but also to other virtual learning environments which focus on medical and public health initiatives [28]. This suggests that these participants find this program is satisfying a previously unmet need.

### 4.2. Facilitated Listening Sessions

The COVID-19 pandemic has caused tremendous social isolation for most people, and yet the participants completing the survey felt less shut down. Whether it is the community of practice, feeling more connected and less detached, the skills they learn, and/or a feeling of hope that they take away from the program, the FRR ECHO sessions appear quite meaningful to the participants. The facilitated listening sessions, which occur at every session, are a novel aspect of this program and may contribute to a feeling of support. These listening sessions likely provide just enough perspective-taking and empathy that participants feel energized to carry on with their duties. Additionally, the sense of connection that they provide in and of itself may buffer the risk of threats to physical and mental health and increase resiliency.

### 4.3. Community of Practice

The FRR ECHO program has provided the TFR and NTFR a supportive community to join synchronously via Zoom each week [29,30]. This program has attracted regular participants as far away as the Philippines, where it is 0500 UTC. Many participants have chosen to attend these FRR ECHO sessions ten or more times, suggesting this program is fulfilling a need for them either psychologically or professionally. Perhaps it is also the freedom to choose to attend (or not) that may be appealing to this community of practice.

These first responders can obtain evidence-based and evidence-guided information related to both self-care and resilience strategies, as well as learn about other nationally regarded programs working to serve frontline healthcare providers. Participants in the focus groups shared that the FRR ECHO provides both a sense of community for them and is of high quality. The Project ECHO model values bi-directionality of learning. Participants are always encouraged to turn on their video camera, and the facilitated listening sessions encourage more intimate discussions [29,30].

### 4.4. Participant Survey

The survey results suggest that the participants (who attended at least four FRR ECHO sessions) were increasing their confidence in skills they have learned in this program. This was despite no change in their environmental stress level. For instance, the survey participants continued to feel fatigued and endorsed that they rarely feel appreciated by the public for their service. In terms of first responders’ ability to cope with their work stress, they had a significant reduction in feeling isolated or “shut down”. It is plausible that the FRR ECHO sessions may help by decreasing moral injury. However, further in-depth investigation is needed to confirm this hypothesis [31].

The survey results also found that many participants noted that they were using alcohol more now than at the beginning of the pandemic. Specifically, on the survey, they marked that “they sometimes drank more alcohol in order to deal with job stress”. Public health officials have hypothesized that alcohol consumption may be increasing during the pandemic, which could cause additional long-term health issues [32].

The FRR ECHO program is providing valuable skills and a needed community of practice to help TFRs and NTFRs increase their confidence during a stressful time, such as the current COVID-19 pandemic. The participants report they are also coping better in many areas except for their alcohol consumption, which has increased nationwide during the pandemic [33].

### 4.5. Focus Group

The focus group participants identified four major themes: (1) external factors are a major cause of stress for participants, (2) the FRR ECHO sessions provide a sense of community, (3) the FRR ECHO sessions are high quality, and (4) participants are sharing the skills they learned in the FRR ECHO with others.

All the themes were identified by NTFR and urban sub-groups of first responders, whereas the rural and TFRs did not experience as much external world stress and felt as though the FRR ECHO course content was less applicable to them (as compared to the NTFR urban sub-groups). The results of the focus group can be explained by the fact that seventy-five percent of the participants were NTFR (and TFR) and live/work in urban settings. Because of this, many of the curriculum topics pertain to the struggles that they faced related to the COVID-19 pandemic. Because the pandemic hit larger cities and suburban areas before rural and remote regions of the country, the course initially focused on issues such as clinician stress and nursing burnout. However, dedicated rural TFR resiliency sessions have also been interspersed through the course.

### 4.6. Predominance of Female Participants

The participant survey and the focus group demographics showed that about twice as many FRR ECHO participants were female. Although Project ECHO does not collect gender information for unique participants of the sessions, it is quite possible that the FRR ECHO is attracting more females than males to the program. If so, this would be in contrast to the predominance of males in the TFR workforce [34]. The workforce for NTFR, on the other hand, is skewed toward more females, with nurses tending to be female in the US, while there are fairly equal numbers of male and female physicians and physician assistants [35]. The implication for the results of the participant survey, which concluded that the subjects learned new skills despite continued work stress but felt less “shut down” after participating in the FRR ECHO, should be taken in the context of this female-predominant survey. The survey also illuminated that alcohol intake increased for all groups of first responders and may relate to the known increase in substance use throughout the US related to pandemic coping skills for both males and females in all first responder categories [32].

It is unclear how the focus group results might have changed for the TFR group had there been more men participating, given the predominantly male workforce among TFRs. The authors realize limitations to the participant survey due to the limited response size. Additional investigation is planned to evaluate substance use among both TFR and NTFR communities, in addition to a deeper investigation into the benefits of the weekly listening sessions.

## 5. Conclusions

The FRR ECHO provides a safe and welcoming place for TFR and NTFR participants during the COVID-19 pandemic. Many participants have chosen to attend this weekly session multiple times, suggesting they benefit from this program. The participant survey and focus group results suggest that participants are developing a supportive community of practice, learning new skills, and sharing them with their colleagues. Additionally, they feel more connected and less detached. Perhaps the weekly FRR ECHO program, providing a combination of the evidence-based didactics and facilitated listening sessions, creates a supportive and revitalizing environment, allowing first responders and frontline clinicians to carry on with their work and home lives through this pandemic with increased resiliency.

## Figures and Tables

**Figure 1 ijerph-18-04900-f001:**
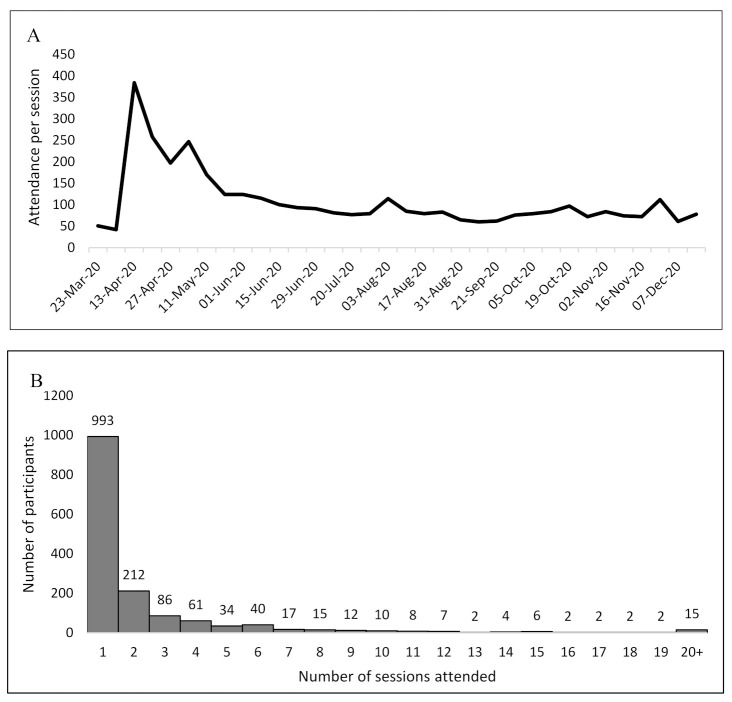
Attendance per session (**A**) and frequency of sessions attended (**B**) of First Responders Resiliency (FRR) ECHO, 23 March 2020–31 December 2020.

**Table 1 ijerph-18-04900-t001:** Characteristics of First Responder Resiliency (FRR) unique participants (31 March–31 December 2020) ECHO survey, and focus groups, 28 September–11 November 2020, N = 45.

	Attendance (N = 919)		Survey (N = 45)		Focus Groups (N = 18)	
**Characteristic**	***n***	**%**	***n***	**%**	***n***	**%**
**Gender**						
Female	-	-	30	66.7%	14	77.8%
Male	-	-	15	33.3%	4	22.2%
**Job Category**						
First Responders (law, fire, EMT)	216	23.5%	16	35.6%	7	38.9%
Medical Professionals (MD, RN, PA)	393	42.8%	16	35.6%	3	16.7%
Mental Health (SW, PsyD, LMFT)	206	22.4%	7	15.6%	6	33.3%
Other Degrees (BS, BA, MPA, PhD)	104	11.3%	6	13.3%	2	11.1%

Many FRR ECHO participants (47.7%) are located in NM. However, others are geographically dispersed throughout all 50 US states and Puerto Rico. Additionally, the FRR ECHO has had attendance from 30 countries—most commonly from Mexico (*n* = 10), South Africa (*n* = 10), Canada (*n* = 7), and Nigeria (*n* = 6). Participants attending from NM represent 28 of NM’s 33 counties. Bachelors Arts (BA), Bachelors Science (BS), Doctor of Philosophy (PhD), Emergency Medical Technician (EMT), Licensed Marriage/Family Therapist (LMFT), Masters in Public Administration (MPA), Medical Doctor (MD), Physician Assistant (PA), Psychologist (PsyD), Registered Nurse (RN), Social Worker (SW).

**Table 2 ijerph-18-04900-t002:** Participants’ confidence ^1^ and agreement ^2^ with statements about resiliency, First Responders Resiliency (FRR) ECHO Survey, 28 September–11 November 2020, N = 45.

Confidence in Using Skills	Mean Score (Before)	Mean Score (Sfter)	*p*-Value
Using the elements of psychological first aid	2.64	3.17	<0.01
Practicing self-care	2.93	3.26	<0.01
Responding to and caring for patients who may be positive for or are suffering from SARS-CoV-2 (COVID-19) ^3^	2.37	2.93	<0.01
Recognizing and managing emergencies related to severe mental illness (e.g., psychosis, depression, etc.) as a first responder	2.64	3.02	<0.01
Incorporating trauma-informed care into your response to emergencies as a first responder	2.40	2.86	<0.01
Recognizing and responding to a coworker struggling with mental health issues	2.77	3.14	<0.01
Recognizing and responding to a coworker struggling with substance use issues	2.56	2.98	<0.01
Recognizing and accessing rural-specific resources to address mental health and substance use in the community	2.57	3.02	<0.01
**Agreement with statements about work stress**			
I often felt tired/fatigued rather than energetic, even when I had enough sleep	3.07	3.02	0.79
I felt overwhelmed by my work	3.12	2.86	0.15
I was absent and sick more often than I’d liked to have been	1.91	1.91	N/A
The public’s lack of respect for my profession was problematic	2.61	2.64	0.89
I often felt in danger while working	2.61	2.66	0.75
**Agreement with statements about coping mechanisms**			
… Spending time with people helped counteract my work stress	3.67	3.84	0.11
… I spent time with various colleagues from work to counteract my work stress	3.43	3.36	0.63
… When I experienced professional stress, I was able to manage it proactively	3.66	3.88	0.15
… When I became stressed due to work, I tended to “shut down” and not talk to others about what is on my mind	2.98	2.65	0.04
... I noticed that my coworkers increased their substance use to deal with job stress	2.93	2.76	0.13
... I sometimes drank more alcohol than I planned to in order to deal with job stress	2.36	3.68	<0.01

^1^ Confidence measured on a 1–4 scale (1-Not at all confident, 2-Somewhat confident, 3-Moderately confident, 4-Extremely confident) ^2^ Agreement measured on a 5-point Likert scale (Strongly disagree, Disagree, Neutral, Agree, Strongly Agree). ^3^ SARS-CoV-2 refers to the COVID-19 virus.

**Table 3 ijerph-18-04900-t003:** Themes and codes in participant feedback by rural versus urban and non-traditional first responder (NTFR) versus traditional first responder (TFR), First Responders Resiliency (FRR) ECHO Focus Groups, 5–9 October 2020, 21 participants.

	Total	Rural		Urban		NTFR		TFR	
**External Factors Caused Stress**	**N ^1^**	***n***	**%**	***n***	**%**	***n***	**%**	***n***	**%**
Experienced world stress	16	4	8.2%	12	9.6%	12	24.0%	4	19.0%
Political and global turmoil	8	2	4.1%	6	4.8%	8	16.0%	0	0.0%
Did not want to be seen as weak	6	2	4.1%	4	3.2%	5	10.0%	1	4.8%
COVID is like a battle	4	3	6.1%	1	0.8%	3	6.0%	4	19.0%
External barriers make skills hard to use	4	1	2.0%	3	2.4%	2	4.0%	2	9.5%
Feeling overwhelmed prevents using skills	1	1	2.0%	0	0.0%	1	2.0%	0	0.0%
**ECHO ^2^ Provides Community**									
Provided sense of community	28	9	18.4%	19	15.2%	16	32.0%	12	57.1%
Reflected/normalized	9	4	8.2%	5	4.0%	6	12.0%	3	14.3%
Encouraged empathy for others	8	3	6.1%	5	4.0%	3	6.0%	5	23.8%
Was a safe space	8	3	6.1%	5	4.0%	3	6.0%	5	23.8%
Found people they can trust	6	0	0.0%	6	4.8%	4	8.0%	2	9.5%
Provided “me” time	3	0	0.0%	3	2.4%	2	4.0%	1	4.8%
Networking opportunities	2	0	0.0%	2	1.6%	0	0.0%	2	9.5%
**Quality of Program**									
Content was applicable	19	5	10.2%	14	11.2%	15	30.0%	4	19.0%
Shared skills with colleagues	13	6	12.2%	7	5.6%	8	16.0%	5	23.8%
Content was a good reminder	7	1	2.0%	6	4.8%	5	10.0%	2	9.5%
Used skills from ECHO	6	1	2.0%	5	4.0%	4	8.0%	2	9.5%
Need more time to learn applications	5	0	0.0%	5	4.0%	5	10.0%	0	0.0%
Not enough time	5	0	0.0%	5	4.0%	4	8.0%	1	4.8%
Zoom format works well	3	0	0.0%	3	2.4%	1	2.0%	2	9.5%
Wanted to be in a breakout with similar people	3	2	4.1%	1	0.8%	3	6.0%	0	0.0%
Liked interdisciplinary teaching	2	1	2.0%	1	0.8%	1	2.0%	1	4.8%
Met participants’ needs	2	1	2.0%	1	0.8%	1	2.0%	1	4.8%
Enjoyed breakout sessions	2	0	0.0%	2	1.6%	2	4.0%	0	0.0%
Content moved too quickly	1	0	0.0%	1	0.8%	1	2.0%	0	0.0%
Participants not wanting to share was a barrier	1	0	0.0%	1	0.8%	0	0.0%	1	4.8%
There were technical problems	1	0	0.0%	1	0.8%	0	0.0%	1	4.8%
Discussion was honest	1	0	0.0%	1	0.8%	0	0.0%	1	4.8%
	174	49	100.0%	125	100.0%	50	100.0%	21	100.0%

^1^ N reflects the number of separate instances where an idea was mentioned, not the number of people, ^2^ ECHO reflects Extension for Community Healthcare Outcomes.

**Table 4 ijerph-18-04900-t004:** Selected participant quotes and corresponding codes, First Responders Resiliency (FRR) ECHO Focus Groups, 5–9 October 2020, 21 participants.

Quote	Codes
*“[It has been helpful] to process different problems that arise whether it’s a tough call or dealing with your coworkers and trying to come in with a clean slate and trying not to come in with that baggage that you have always. That interfere[s] with working with other people and working with patients.”*	Content was applicable
*“…as a healthcare worker … you don’t want to be the one seen as weak, so you have to have a safe space to go to where you can … process everything that you’re going through.”*	Did not want to be seen as weak, Was a safe space
*“…just sharing feelings with other people from different states. They’re going through the same thing, so I’m not alone.”*	Provided sense of community
*“in law enforcement, sometimes we have just one-way vision, and I think that with everything that’s going on now for law enforcement and the climate right now, I think it’s been really important for all of us [to have] compassion towards each other and those stressors and with how we approach our jobs.”*	Encouraged empathy for others
*“I thought that the material that we got was really helpful, and I think that the listening groups offered a little bit of time to talk about some of those a little bit and kind of apply them.”*	Content was applicable, Enjoyed breakout sessions
*“One of the things that … I really like about this program is often times when I am working with other individuals after we’ve had a call or whatever is going on is that it just gives me more things to be able to turn to and set up for them.”*	Shared skills with colleagues

## Data Availability

The data presented in this study are available on request from the corresponding author. The data are not publicly available as it is not standard practice to publicly share protected and confidential research data. Should data sharing be requested, a Data Use Agreement (DUA) between the University of New Mexico and the requesting organization/institution is required.

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
