# Peer review of "First Responder Resiliency ECHO: Innovative Telementoring during the COVID-19 Pandemic"

_ijerph, 2021, doi:10.3390/ijerph18094900_

Round 1

Reviewer 1 Report

The introduction is clear but it does not present what is already known about the topic; this information is mentioned too briefly in the introduction and had better be extended and enriched. Methods are described clearly and an appropriate design is applied. On the other hand, more information is needed regarding the tool the authors used for the online survey (description of the development process). The results are not presented in detail and it is suggested to present the qualitative results more extensively.  Add at the beginning of the discussion one or two paragraphs with the main results of your study. In general, it is a very interesting manuscript that needs more attention to the aforementioned issues.

Author Response

Response to reviewers- IJERPH submission 1150001- First Responder Resiliency ECHO: Innovative Telementoring during the COVID-19 Pandemic

Reviewer 1 comments

Page/line numbers

Author responses

1

The introduction can be improved to provide sufficient background and include all relevant references

Pgs 1-2, lines 50-98 & 104-106

We thank the reviewers for the feedback. The authors have added much additional background information (and references) related to both the impact of first responder psychological trauma as well as some of the research known about first responder resiliency.

We are requesting to add a co-author, Karina Dow, to our current article. Ms. Dow has performed much of the additional research for this article. She had been planning to submit a separate manuscript however, we now plan to forgo that in order to meet the needs of this manuscript.

2

The presentation of the results must be improved for clarity

Pgs 4-6, lines 195-259

The results have been reordered and formatted for clarity and to meet the journal’s formatting guidelines for the results section

3

The introduction is clear but it does not present what is already known about the topic; this information is mentioned too briefly in the introduction and had better be extended and enriched.

Pgs 1-2, lines 50-98 & 104-106

The authors have added a significant amount of new information about previous first responder resiliency research investigations.

4

Methods are described clearly and an appropriate design is applied. On the other hand, more information is needed regarding the tool the authors used for the online survey (description of the development process).

Pg 4, lines 157-169

The authors have added additional information describing the development process of the online survey.

5

The results are not presented in detail and it is suggested to present the qualitative results more extensively

Pgs 5-6, lines 237-259; and Table 4

The authors have expanded on the description of the qualitative analysis.

6

Add at the beginning of the discussion one or two paragraphs with the main results of your study.

Pg 6, lines 262-273

The authors have added two detailed paragraphs to the beginning of the discussion section that proved a succinct, but inclusive overview of the most salient conclusions in from this study.

Author responses

Reviewer 2 Report

I enjoyed reading this paper..most of the parts - methods..discussions..conclusions, etc are very well laid out. This only aspect I would suggest improvement is with the use of the concept of resilience. As someone that has done work tryingto develop the concept/theory of resilience with regards to the problems Africans face in China and how they react to them, I was a bit disappointed that resilience was only just mentioned in the title and hardly ever discussed in the body of the work. 

Author Response

Reviewer 2 comments

Page/line numbers

Author responses

The introduction can be improved to provide sufficient background and include all relevant references

Pgs 1-2, lines 50-98 & 104-106

The authors have added a significant amount of new information about previous first responder resiliency research investigations

The only aspect I would suggest improvement is with the use of the concept of resilience. As someone that has done work trying to develop the concept/theory of resilience with regards to the problems Africans face in China and how they react to them, I was a bit disappointed that resilience was only just mentioned in the title and hardly ever discussed in the body of the work.

Pgs 1-2, lines 50-98 & 104-106

Pg 2, lines 96-98

Pg 6, lines 272-273

Pg 6, 296-298

Pg 8, lines 374-378

The authors are grateful for this critique and agree with the lack of discussion around resilience in the article- despite the fact that the program emphasizes many aspects of resiliency.

We relooked at all sections and aspects of the article. We added much resiliency research. In addition, we discussed the relationship between social support, listening, empathy and perspective taking with many aspects of resiliency.

Round 2

Reviewer 1 Report

The updated version of the manuscript is based on the valuable comments of the reviewers. I think that the interest for the readers will be high because of the scientific soundness and the significance of the content as well.